# Pre-treatment subjective sleep quality as a predictive biomarker of tDCS effects in preclinical Alzheimer's disease patients: Secondary analysis of a randomised clinical trial

**Hanna Lu** [1,2]*, **Xi Ni**[1], **Sandra Sau Man Chan**[1], **Calvin Pak Wing Cheng** [3], **Waichi Chan**[3], **Linda Chiu Wa Lam**[1]

**1** Department of Psychiatry, The Chinese University of Hong Kong, Sha Tin, Hong Kong SAR, China, **2** The Affiliated Brain Hospital of Guangzhou Medical University, Guangzhou, China, **3** Department of Psychiatry, The University of Hong Kong, Pokfulam, Hong Kong SAR, China

* hannalu@cuhk.edu.hk

## Abstract

### Background

Despite transcranial direct current stimulation (tDCS) has demonstrated encouraging potential for modulating the circadian rhythm, little is known about how well and sustainably tDCS might improve the subjective sleep quality in older adults. This study sought to determine how tDCS affected sleep quality and cognition, as well as how well pretreatment sleep quality predicted tDCS effects on domain-specific cognitive functions in patients with mild neurocognitive disorder due to Alzheimer's disease (NCD-AD).

### Methods

This clinical trial aimed to compare the effectiveness of tDCS and cognitive training in mild NCD-AD patients (n = 201). Over the course of four weeks, patients were randomized to receive either tDCS plus working memory training, or sham tDCS plus working memory training, or tDCS plus controlled cognitive training. The Pittsburgh Sleep Quality Index (PSQI) was used to measured subjective sleep quality. The Alzheimer's disease assessment scale-cognitive subscale (ADAS-Cog) was used to evaluate domain-specific cognitive functions.

### Results

Recurrent tDCS treatments enhanced subjective sleep quality and cognition considerably. The poor sleepers (i.e., PSQI > 5) who received tDCS treatment had more cognitive benefits ($p = 0.031$, Cohen's $d = 0.605$) and sleep improvements ($p < 0.001$, Cohen's $d = 1.209$) in comparison to cognitive training. Pre-treatment subjective sleep quality was linked to tDCS-induced improvement in memory function.

**Data availability statement:** This study has the potential to identify cases by clinical information, and has received guidance on strict controls from the Joint Chinese University of Hong Kong-New Territories East Cluster Clinical Research Ethics Committee in Japan. Therefore, this study has ethical constraints on the sharing of the datasets. However, researchers who meet the criteria for access to confidential data, may submit a collaboration request to Ms. Yuk Shan Yuen via yukshanyuen@cuhk.edu.hk, including the formal information on the institution and a brief description of the project.

**Funding:** This clinical trial was supported by the Hong Kong Research Grant Council (RGC) - General Research Fund (GRF) (GRF14108214). The funders had no role in study design, data collection and analysis, decision to publish, or preparation of the manuscript.

**Competing interests:** The authors have no conflicts of interests to declare.

## Conclusion

During the course of two months, repeated tDCS could considerably enhance subjective sleep quality. For the cognitive benefits of the treatments, the status of pre-treatment subjective sleep quality is crucial. More thorough research is necessary to explore an efficient approach to managing comorbidities for preclinical AD patients.

## Introduction

In the processes of healthy ageing and neurodegeneration, subjective sleep quality is increasingly being described as an emerging feature of brain health [1,2]. One of the main reasons is that in late adulthood, sleep undergoes significant alternations in its structure and endogenous circadian biology [3], which are linked to neurovascular diseases, rapid cognitive decline, and poor prognosis [1,4]. Because of the reciprocal relationships between sleep quality and cognitive functioning, poor sleep quality may exacerbate the clinical status of cognitive impairments [5]. Older adults who experience poor sleep quality are also more likely to have poorer cognitive performance and even be at higher risk of developing neurodegenerative diseases [6–9]. Thus, the growing interest in the connections between subjective sleep quality and cognitive functions is driving the development of innovative therapies for the successful management of these two comorbidities in older adults.

Non-pharmacological treatments for enhancing sleep quality are accepted for use in patients with dementia; however, it is still unclear whether these treatments are effective and long-lasting in improving subjective sleep quality and cognitive abilities in preclinical Alzheimer's disease (AD). Transcranial direct current stimulation (tDCS) is a user-friendly, non-invasive neurotechnology that modulates the neural excitability by applying mild electric currents to the head, usually between 1 and 2 milliamperes [4]. Preliminary research has indicated that tDCS may enhance older adults' cognitive performance and alter their circadian rhythm [10]. There is growing evidence that tDCS can improve subjective sleep quality in a variety of demographic groups, including normal ageing adults [11], young adults with sleep deprivation [12–14], senior adults with healthy ageing [15,16], patients with mood disorders [17,18] and migraine [19]. According to a recent study, repeated tDCS may have greater impacts on older adults' subjective sleep quality than the those between the ages of 18 and 50 [20]. Despite the fact that tDCS's effects on sleep have been extensively studied, little is known about how it impacts domain-specific cognitive functions or how subjective sleep quality predicts cognitive changes in randomized clinical trials.

At present, no published investigation has reported that repeated sessions of anodal tDCS can modulate subjective sleep quality and domain-specific cognitive functions in a sizable sample of old adults with mild neurocognitive disorder due to Alzheimer's disease (NCD-AD). Therefore, our primary aim was to provide a comprehensive picture of subjective sleep quality, global cognition, and domain-specific cognitive functions related to tDCS treatments in order to significantly advance our understanding of the reciprocal relationships between subjective sleep quality and cognitive functions. Finding connections between pre-treatment subjective sleep quality and treatment-induced cognitive changes was the secondary aim of this study.

## Materials and methods

### Study participants

The study is a secondary analysis of a randomized clinical trial (RCT) [21]. Study methods and participants characteristics were listed in the Consolidated Standards for Reporting

Trials (CONSORT) flow diagram (Fig 1). Additional details of study design can be found in the published research protocol [22]. Briefly, from May 4, 2015 to November 30, 2017, six hundred and forty-three adult participants were screened, 201 right-handed Chinese adults aged from 60 to 90 years with a diagnosis of mild neurocognitive disorder due to Alzheimer's disease (NCD-AD) were recruited in this study. The following inclusion criteria were used to define mild NCD-AD participants: (1) evidence of modest cognitive decline in one or more cognitive domains, as indicated by a box score of ≤ 0.5 on the Clinical Dementia Rating (CDR) subcategory and a Cantonese Mini Mental State Examination (CMMSE) score ranging from 22 to 27; (2) no interference with daily activities that require independence; (3) with a score of episodic memory as measured by delay recall in list learning; and (4) no better explanation by other psychiatric disorders. In addition to meeting the criteria for NCD, patients with mild NCD-AD did not have uncontrolled hypertension, diabetes, or hyperlipidemia (vascular risk factors). The exclusion criteria were as follows: (1) previous diagnosis of dementia or other major neurocognitive disorders; (2) past history of bipolar affective disorder or psychosis; (3) history of major neurological deficit including stroke, transient ischemic attack or traumatic brain injury; (4) taking a psychotropic or other medication known to affect cognition (e.g., benzodiazepines, antidementia medication, etc.); (5) physically frail affecting attendance to

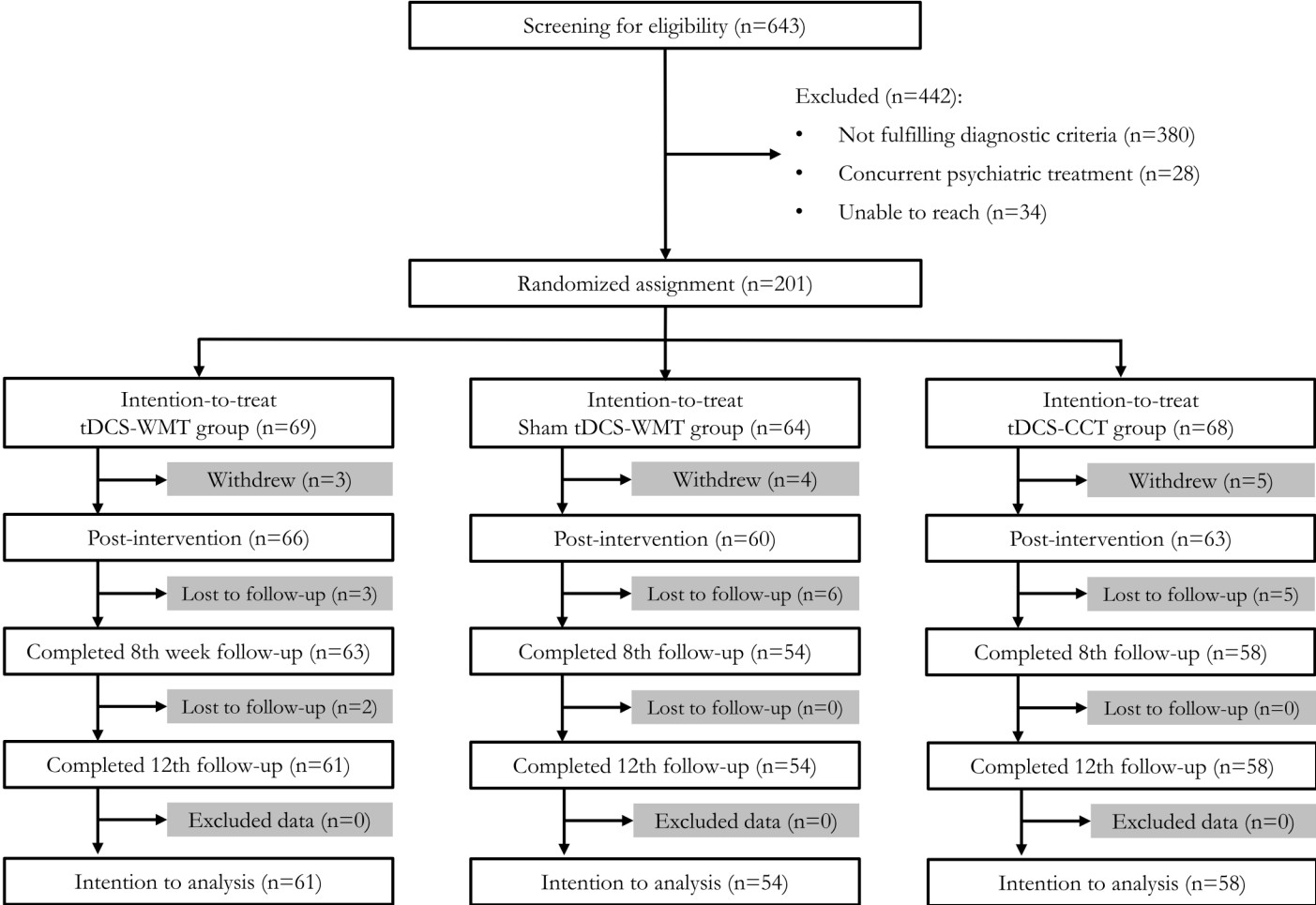

**Fig 1. The Consolidated Standards of Reporting Trials (CONSORT) flow diagram for transcranial direct current stimulation (tDCS) clinical trial.**

training sessions; (6) already attending regular cognitive training; and (7) significant communication problems.

A randomization sequence was generated to allocate 69 participants to the tDCS-working memory training (tDCS-WMT) group, 64 participants in the sham tDCS-working memory training (sham tDCS-WMT) group, and 68 participants in the tDCS-controlled cognitive training (tDCS-CCT) group. Eligible participants were scheduled for a 4-week course of treatment and four times of assessments at baseline, 4th week, 8th week and 12th week. The Joint Chinese University of Hong Kong-New Territories East Cluster Clinical Research Ethics Committee had approved this clinical trial. The written informed consent was obtained from all participants or their legal guardian(s) before any assessment. This study was conducted in accordance with the declaration of Helsinki and relevant guidelines and regulations.

## Assessments of sleep quality and cognition

The Pittsburgh Sleep Quality Index (PSQI) is a 19-item self-rated questionnaire that measures sleep quality and the severity of sleep disturbances along seven components (C) [23]: C1: subjective sleep quality; C2: sleep latency; C3: sleep duration; C4: habitual sleep efficiency; C5: sleep disturbance; C6: use of sleeping medications; C7: daytime dysfunction over the last month. Higher score of PSQI represents worse subjective sleep quality and severe sleep disturbances.

The Alzheimer's disease assessment scale-cognitive subscale (ADAS-Cog), as an assessment of global cognition, comprises ratings from eleven items, including word recall, word recognition, constructional praxis, orientation, naming objects and fingers, commands, ideational praxis, remembering test instructions, spoken language, word finding, and comprehension [24]. Based on the psychometric features, the ADAS-Cog covers four main cognitive domains: memory, language, praxis and orientation [25]. The total score of ADAS-Cog ranges from 0 to 70, with increasing total score indicating worse global cognition.

## Safety assessments

The Adverse Event Checklist (AEC) covers the symptoms or problems of eight system [26], including body as whole, cardiovascular system, digestive system, hemic and lymphatic system, metabolic and nutritional systems, musculoskeletal system, nervous system and urogenital system.

## Statistical analyses

Analyses were performed by the "intention-to-treat" principles. The between-group comparisons of continuous variables were conducted using multilevel generalized linear mixed models. Time was treated as a categorical variable. The models included group, time, and group by time interaction as fixed effect and participants as random effect. Data is expressed as score change from baseline to the follow-up time points, determined by the time coefficients (95% confidence interval [CI]) of the model. For each group, score changes of subjective sleep quality and cognitive functions from baseline to each follow-up point and treatment group differences were tested with occasions (time points) at level one and participants at level two. Secondary analyses of specific cognitive domains were performed to compute for group differences in cognitive outcomes. All comparisons were two-sided, with a statistical significance level of 0.05. The computations were performed using R package.

A correlation matrix was computed between the score changes of sleep quality and domain-specific cognitive functions, age, education and baseline sleep quality in each randomized group. The score changes were primarily calculated based on the raw scores at

baseline and follow-up time points. Baseline PSQI total scores were included as predictors for the score changes of global cognitive function. Correlation with age and the years of education was also determined. Visualization of the correlation matrix of data across the participants in each group was done using the *Corrplot* package in R (https://github.com/taiyun/corrplot).

## Results

### Patient population

Of the 201 included in the analyses, the mean age was 73.8 years (age range: 60.5–90.9 years); the mean PSQI total score was 5.8 (ranged from 0 to 15). Demographic information, subjective sleep quality and clinical characteristics were listed in Table 1. Since twenty-eight participants dropped out from this study (dropout rate: 13.9%), a total number of 173 participants were included in the intention-to-analysis.

### Outcomes of subjective sleep quality

In general, 41.3% (83/201) of the mild NCD-AD patients had sleep disturbances with a cut-off PSQI total score of 5 [27]. After a 4-week treatment, heterogeneous treatment responses were observed in PSQI total score at all follow-up time points. At week 12, the patients who received tDCS-CCT treatment showed a noticeable improvement in their sleep quality (Score change: $F = 9.5$, $p = 0.003$, Cohen's $d = 0.89$) when compared to those who received tDCS-WMT and sham tDCS-WMT treatments.

In order to further examine how pre-treatment subjective sleep quality statuses affect treatment-induced cognitive changes, we divided the mild NCD-AD patients into baseline good sleepers and poor sleepers for each treatment group. We defined good sleepers as PSQI total score ≤ 5, and poor sleepers as baseline PSQI total score > 5. There were no baseline differences between good sleepers and poor sleepers in either demographic information or global cognition (S1 Table). After the treatments, the poor sleepers showed significantly improved subjective sleep quality than the good sleepers (S1 Fig). During the follow-up time periods, the changes of the seven PSQI components were also observed in three groups (S2 Fig

**Table 1. Demographics, sleep quality and neurocognitive characteristics across the randomized groups.**

|  | tDCS-WMT (n = 69) | Sham tDCS-WMT (n = 64) | tDCS-CCT (n = 68) | *F Value* | *P Value* |
|---|---|---|---|---|---|
| Age | 74.21 ± 6.65 | 74.45 ± 6.59 | 73.42 ± 6.12 | 0.47 | 0.63 |
| Sex (F/M) | 42/21 | 36/17 | 30/27 | 2.35 | 0.09 |
| Years of education | 7.29 ± 4.82 | 6.48 ± 4.29 | 7.51 ± 5.28 | 0.81 | 0.45 |
| PSQI | 5.68 ± 3.66 | 5.11 ± 3.33 | 6.15 ± 3.38 | 1.48 | 0.23 |
| CDR | 1.09 ± 0.74 | 1.23 ± 0.96 | 1.19 ± 0.66 | 0.54 | 0.58 |
| CMMSE | 25.65 ± 2.64 | 25.64 ± 2.89 | 25.47 ± 2.50 | 0.10 | 0.91 |
| ADAS-Cog (total) | 9.35 ± 3.91 | 9.35 ± 3.96 | 9.71 ± 3.95 | 0.28 | 0.76 |
| Memory | 11.63 ± 2.21 | 11.50 ± 2.14 | 11.95 ± 2.28 | 0.61 | 0.55 |
| Language | 0.57 ± 0.91 | 0.74 ± 0.90 | 0.68 ± 0.93 | 0.49 | 0.61 |
| Praxis | 0.57 ± 0.91 | 0.74 ± 0.90 | 0.68 ± 0.93 | 0.54 | 0.58 |
| Orientation | 11.63 ± 2.21 | 11.50 ± 2.14 | 11.95 ± 2.28 | 0.17 | 0.84 |

Note. Data are raw scores and presented as mean ± SD.

Abbreviations: tDCS = Transcranial direct current stimulation; WMT = Working memory training; CCT = Controlled cognitive training; PSQI = Pittsburgh Sleep Quality Index; CDR = Clinical Dementia Rating; CMMSE = Cantonese Mini-Mental State Examination; ADAS-Cog = The Alzheimer's Disease Assessment Scale-Cognitive Subscale.

and S2 Table ). Within sham tDCS-WMT group, the poor sleepers had more improvements in habitual sleep efficiency than the good sleepers at 4th week (C4: $t = -2.538$, $p = 0.018$) and 12th week (C4: $t = -2.259$, $p = 0.033$). Within tDCS-CCT group, the poor sleepers had more improvements in sleep latency (C2: $t = -2.651$, $p = 0.011$), sleep disturbance (C5: $t = -2.305$, $p = 0.025$) and use of sleeping medications (C6: $t = -3.132$, $p = 0.003$) than the good sleepers at 12th week (Table 2).

## Outcomes of cognitive functions

Global cognition and domain-specific cognitive functions were analyzed as the score changes from baseline to post-treatment follow up time points. Across all follow-up observations, the poor sleepers had generally benefited more from global cognition than the good sleepers (Score changes of ADAS-Cog: 4th week, $t = -2.42$, $p = 0.019$, Cohen's $d = 0.669$; 8th week, $t = -2.19$, $p = 0.031$, Cohen's $d = 0.605$) (Fig 2). Throughout the follow-up observations, the poor sleepers exhibited greater cognitive improvements on the domains of memory, language, praxis, and orientation compared to the good sleepers (Fig 3). Within tDCS-CCT group, significant improvements in memory function were seen in the poor sleepers at 4th week ($t = -2.152$, $p = 0.036$) and 12th week ($t = -3.162$, $p = 0.003$).

## Adverse events and safety

All patients reported that the treatments were well-tolerated following twelve sessions of treatments. In tDCS groups, the most significant adverse events were mild skin injury under

**Table 2. Comparisons of sleep components in mild NCD-AD with statuses of subjective sleep quality (score changes).**

| Cognitive features | tDCS-WMT (n = 62) | | | | Sham tDCS-WMT (n = 53) | | | | tDCS-CCT (n = 57) | | | |
|---|---|---|---|---|---|---|---|---|---|---|---|---|
| | Good sleepers | Poor sleepers | t value | p value | Good sleepers | Poor sleepers | t value | p value | Good sleepers | Poor sleepers | t value | p value |
| Component 1 | | | | | | | | | | | | |
| 4th week | −0.21 ± 0.86 | 0.36 ± 1.44 | −1.895 | 0.063 | −0.12 ± 0.82 | −0.11 ± 1.25 | −0.075 | 0.941 | 0.01 ± 0.77 | 0.11 ± 1.32 | −0.408 | 0.685 |
| 12th week | −0.24 ± 0.96 | 0.15 ± 1.13 | −1.439 | 0.156 | −0.24 ± 0.91 | −0.15 ± 0.87 | −0.365 | 0.716 | −0.09 ± 0.99 | 0.43 ± 1.17 | −1.712 | 0.093 |
| Component 2 | | | | | | | | | | | | |
| 4th week | 0.01 ± 0.64 | 0.18 ± 0.47 | −1.227 | 0.225 | 0.01 ± 0.66 | 0.05 ± 0.51 | −0.289 | 0.773 | −0.09 ± 0.71 | 0.29 ± 0.71 | −1.954 | 0.056 |
| 12th week | −0.09 ± 0.52 | 0.11 ± 0.51 | −1.511 | 0.136 | −0.21 ± 0.55 | 0.01 ± 0.46 | −1.518 | 0.136 | −0.24 ± 0.71 | 0.23 ± 0.59 | **−2.651** | **0.011** |
| Component 3 | | | | | | | | | | | | |
| 4th week | 0.01 ± 0.87 | 0.29 ± 1.35 | −1.011 | 0.316 | 0.06 ± 0.75 | 0.45 ± 0.88 | −1.713 | 0.093 | −0.14 ± 0.85 | 0.31 ± 1.18 | −1.544 | 0.128 |
| 12th week | −0.03 ± 1.01 | 0.37 ± 1.11 | −1.456 | 0.151 | −0.24 ± 0.79 | 0.15 ± 1.09 | −1.515 | 0.136 | −0.05 ± 0.92 | 0.34 ± 1.03 | −1.431 | 0.158 |
| Component 4 | | | | | | | | | | | | |
| 4th week | −0.17 ± 0.71 | 0.18 ± 0.86 | −1.771 | 0.082 | −0.06 ± 0.35 | 0.35 ± 0.67 | **−2.538** | **0.018** | −0.05 ± 0.74 | 0.31 ± 0.76 | −1.745 | 0.087 |
| 12th week | −0.09 ± 0.63 | 0.15 ± 0.77 | −1.323 | 0.191 | −0.12 ± 0.41 | 0.35 ± 0.87 | **−2.259** | **0.033** | −0.14 ± 0.91 | 0.31 ± 0.87 | −1.875 | 0.066 |
| Component 5 | | | | | | | | | | | | |
| 4th week | −0.11 ± 0.76 | 0.32 ± 1.51 | −1.485 | 0.143 | −0.27 ± 0.72 | −0.11 ± 1.55 | −0.551 | 0.584 | −0.19 ± 0.87 | 0.41 ± 1.73 | −1.688 | 0.097 |
| 12th week | −0.45 ± 1.25 | −0.11 ± 1.57 | −0.941 | 0.351 | −0.58 ± 1.01 | −0.11 ± 1.16 | −1.576 | 0.121 | −0.29 ± 1.19 | 0.61 ± 1.49 | **−2.305** | **0.025** |
| Component 6 | | | | | | | | | | | | |
| 4th week | −0.03 ± 0.67 | 0.04 ± 0.83 | −0.339 | 0.736 | 0.06 ± 0.61 | 0.11 ± 0.72 | −0.213 | 0.832 | −0.05 ± 0.86 | 0.14 ± 0.61 | −0.972 | 0.336 |
| 12th week | −0.03 ± 0.64 | 0.19 ± 0.68 | −1.264 | 0.211 | −0.09 ± 0.68 | 0.05 ± 0.83 | −0.675 | 0.503 | −0.19 ± 0.61 | 0.29 ± 0.52 | **−3.132** | **0.003** |
| Component 7 | | | | | | | | | | | | |
| 4th week | 0.06 ± 0.34 | −0.11 ± 0.41 | 1.729 | 0.089 | 0.01 ± 0.01 | 0.11 ± 0.72 | −0.805 | 0.425 | 0.01 ± 0.01 | 0.03 ± 0.45 | −0.288 | 0.774 |
| 12th week | 0.03 ± 0.39 | 0.01 ± 0.28 | 0.337 | 0.737 | 0.01 ± 0.01 | 0.01 ± 0.32 | 0.001 | 1.000 | 0.01 ± 0.01 | 0.09 ± 0.45 | −0.139 | 0.263 |

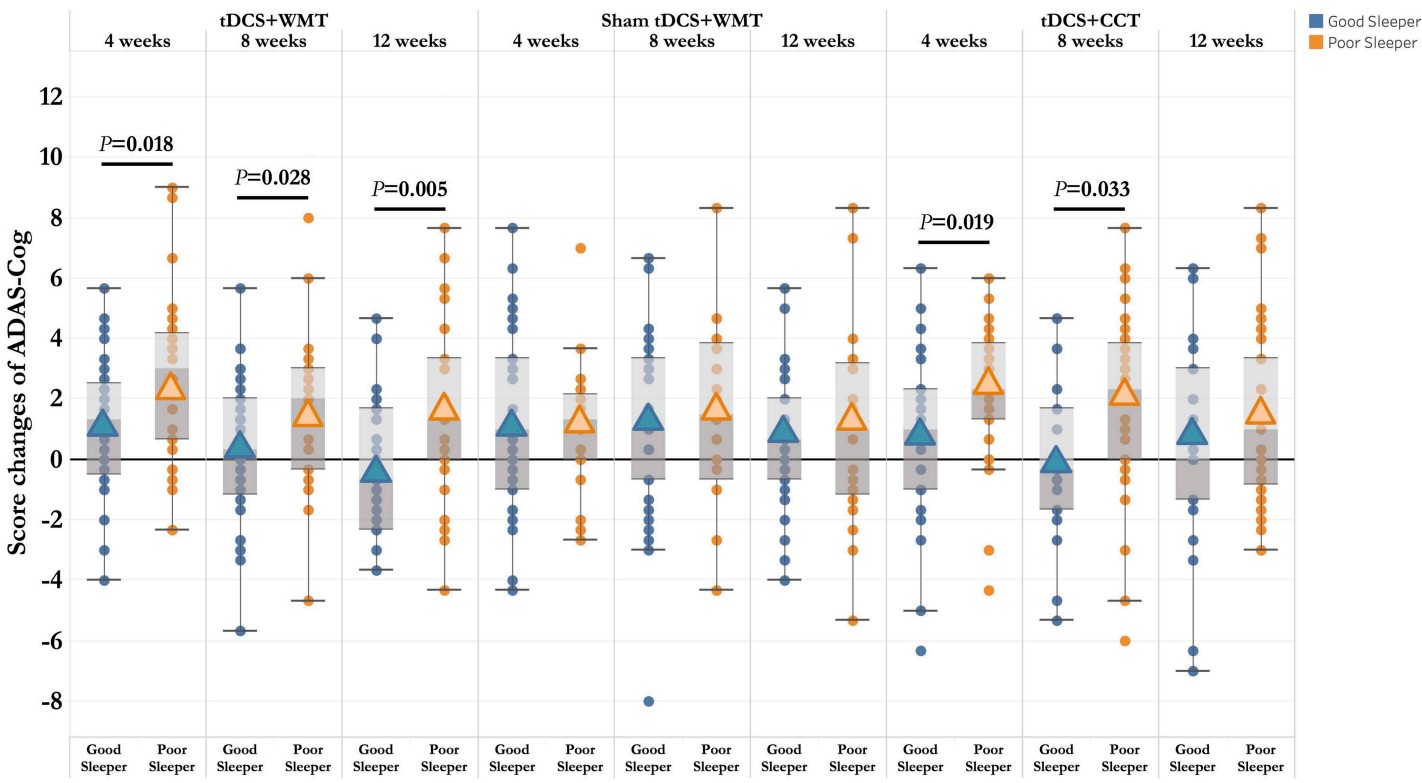

**Fig 2. Comparisons of the treatment-induced changes of ADAS-Cog across follow-up observations.** The poor sleepers had more cognitive gains than the good sleeps. Box plot represents the differences of ADAS-Cog score changes across time points. Abbreviations: tDCS = Transcranial direct current stimulation; WMT = Working memory training; CCT = Controlled cognitive training; ADAS-Cog = The Alzheimer's Disease Assessment Scale–Cognitive Subscale.

cathodal electrode in three patients [26]. However, the above skin injuries were recovered after one week. None of the patients reported other adverse events, such as nausea, seizures, headache, or dizziness.

## Correlation matrices

In general, baseline PSQI total score was significantly correlated with memory function ($r =$ 0.225, $p = $ 0.003), indicating that poor sleep quality was related to the worse performance on memory in mild NCD-AD patients.

In the treatment groups, three correlation matrices were calculated to better understand the relationships between baseline sleep quality, age, years of education, and the changes in both sleep quality and cognitive function. Overall, using sex as covariate, there was a substantial correlation between baseline PSQI score and changes in PSQI scores over the duration of the follow-up time points. Improved sleep quality following the treatments was significantly correlated with lower baseline sleep quality. The tDCS-WMT and tDCS-CCT groups had higher Pearson correlation coefficients than the sham tDCS-WMT group. However, there were persistent and significant correlations between the baseline PSQI score, years of education, and the changes in ADAS-Cog scores only in the tDCS-WMT (Fig 4a) and tDCS-CCT (Fig 4c) groups, not the sham tDCS-WMT group (Fig 4b). In tDCS-WMT and tDCS-CCT groups, less years of education was correlated with higher baseline PSQI total score (i.e., poor sleep quality) (tDCS-WMT group: $r = $ -0.292, $p = $ 0.031; tDCS-CCT group: $r = $ -0.276, $p = $ 0.037). Poorer

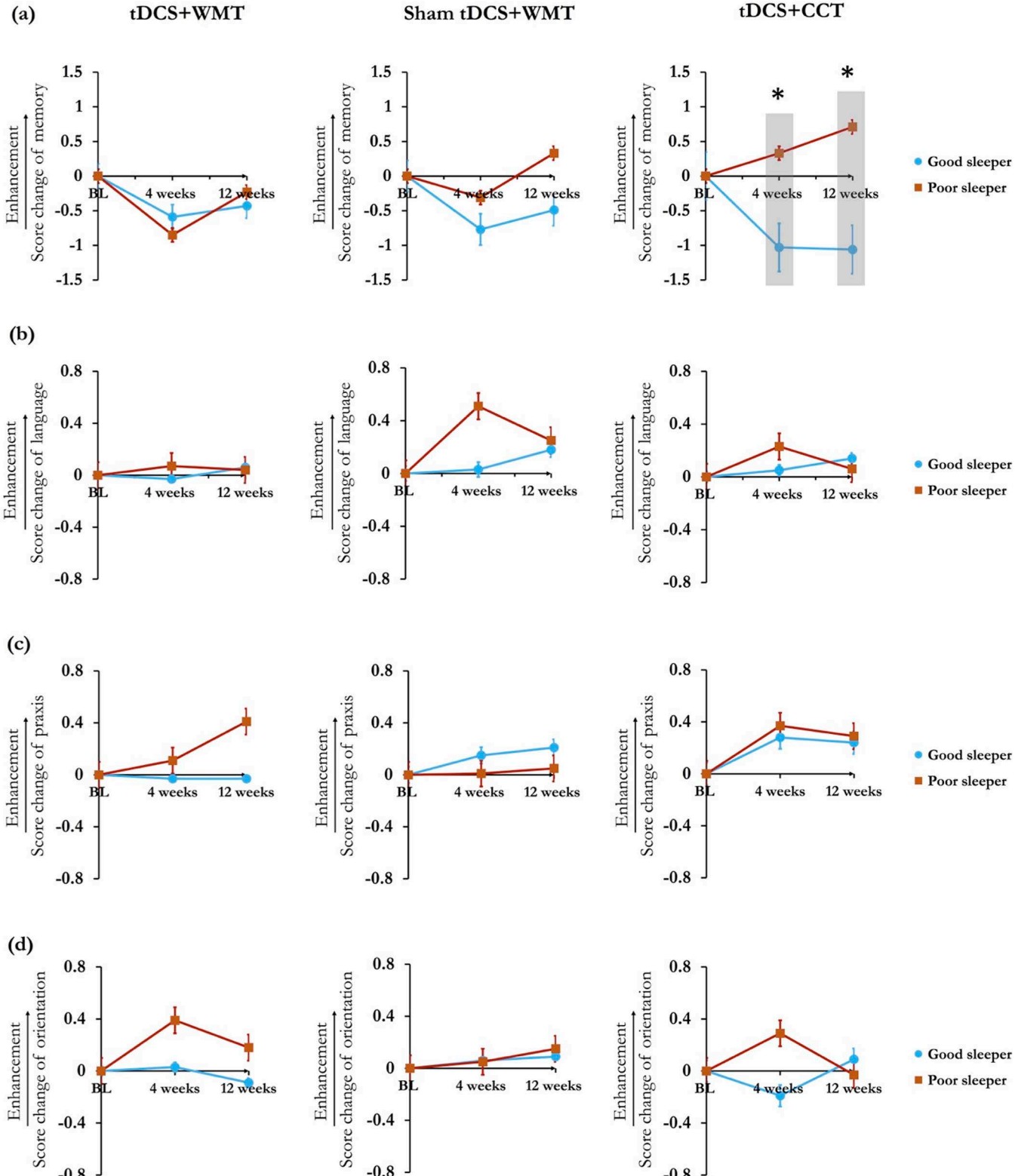

**Fig 3. Comparisons of the treatment-induced changes of domain-specific cognitive functions, including (a) memory (b) language (c) praxis, and (d) orientation.** In the individuals received tDCS-CCT treatment, the poor sleepers had more memory enhancement than the good sleeps. Blue line represents the good sleeper; red line represents poor sleeper. Abbreviations: tDCS = Transcranial direct current stimulation; WMT = Working memory training; CCT = Controlled cognitive training. Error bars represent the standard error (SEM). * Indicates significant between-group difference ($p < 0.05$).

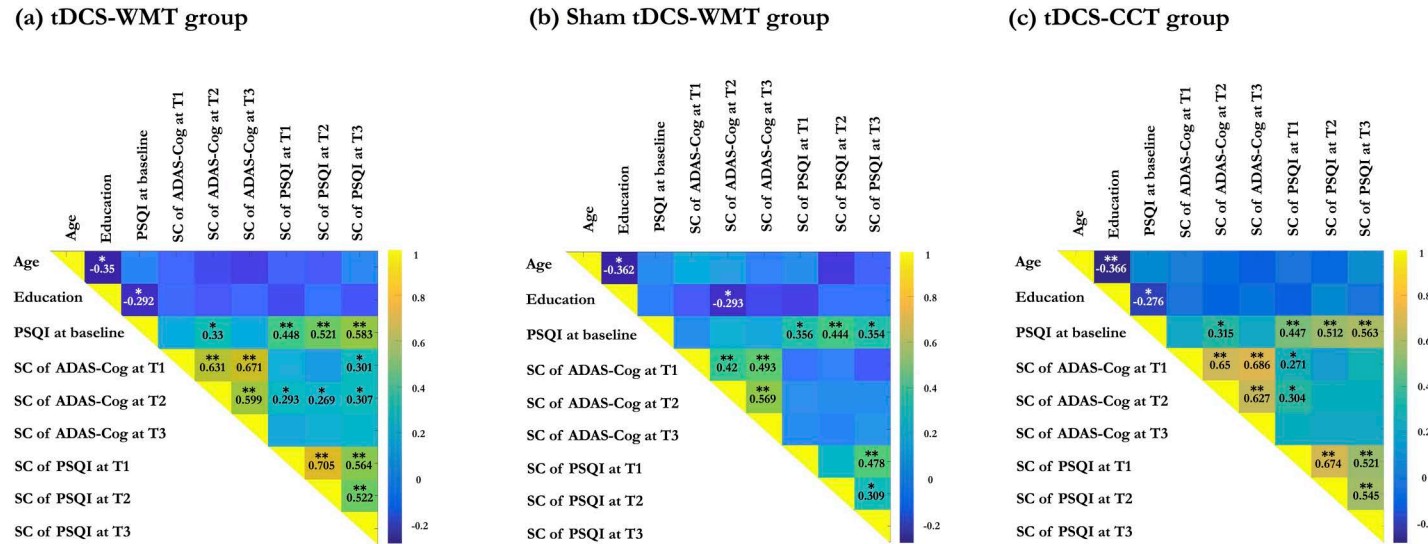

**Fig 4. Correlation matrix for the changes in sleep quality and global cognitive function.** The colour scale depicts the strength of the Pearson correlation coefficient. Superimposed text represents the actual numerical value of the Pearson correlation coefficient. The asterisk depicts the statistical significance of the correlation: * represents $p < 0.05$, ** represents $p < 0.005$. Abbreviations: tDCS = Transcranial direct current stimulation; WMT = Working memory training; CCT = Controlled cognitive training; PSQI = Pittsburgh Sleep Quality Index; SC = Score change; ADAS-Cog = The Alzheimer's Disease Assessment Scale–Cognitive Subscale.

sleep quality at baseline was substantially correlated with more global cognitive gains at 8th week (tDCS-WMT group: $r = 0.33$, $p = 0.014$; tDCS-CCT group: $r = 0.315$, $p = 0.018$).

## Discussion

The primary aim of this secondary analysis of a clinical trial was to provide a comprehensive picture of subjective sleep quality, global cognition, and domain-specific cognitive functions related to tDCS treatments in mild NCD-AD patients. Our results point to the possibility that repeated anodal tDCS treatments over the left temporal cortex might improve subjective sleep quality and offer greater advantages than cognitive training. Pre-treatment subjective sleep quality may be a significant predictor of the results of tDCS treatment. This is the first clinical study showing that repeated tDCS and cognitive training treatments improve subjective sleep quality, global cognition and the functions of main cognitive domains in mild NCD-AD patients.

### TDCS effects on sleep quality

In our study, 41.3% of the mild NCD-AD patients experienced sleep disturbances or poor sleep quality, which is consistent with earlier studies that found poor sleep quality affects more than 50% of mild-to-moderate AD patients at advanced stages [28]. The high incidence of sleep disturbances and the associated decline in memory function that we found in this study raises the difficulties of appropriately treating the comorbidity in people at the early stage of Alzheimer's disease. Among the intriguing results is the synergistic effects of tDCS and cognitive training on subjective sleep quality. While comparable synergistic effects on cognitive functions have been observed [21], only the those who received tDCS treatments (i.e., tDCS-WMT and tDCS-CCT) showed a substantial improvement in their sleep quality. At the endpoint, the tDCS-CCT group outperformed the tDCS-WMT group in terms of overall improvements in subjective sleep quality.

As a controlled condition of working memory, the controlled cognitive training (CCT) is a computerized test of focused attention (i.e., processing speed). Either working memory training or CCT, the core components of these training are the domains of cognitive functions that are working memory and sustained attention. The observed interactions between tDCS and cognitive training in old adults may explain the differential results. For example, the involvement of attention function as a component of the attention training (i.e., CCT) might be the reason for greater enhancement in subjective sleep quality in the tDCS-CCT group. An emerging theory to explain these benefits is that attention training modulates the activities of the ascending reticular activating system (ARAS), such as sleep-alertness pathway. Nonetheless, sleep disturbances are common chronic diseases among elderly patients that are lined to a higher risk of accelerated cognitive decline. Discrepancies across published studies may be attributable, at least in part, to methodological differences, because the majority of tDCS studies have been conducted in a smaller sample of non-clinical groups, such as young adults [29]. Alternatively, research using older adults without cognitive deficits [30] might be unable to document the cognitive changes linked to the enhancements in sleep quality following the treatments.

## The role of sleep quality in predicting tDCS effects

For mild NCD-AD patients with or without sleep disturbances, recurrent anodal tDCS appears to be an effective treatment to improve subjective sleep quality, which is critical for older adults' brain health and quality of life. Recent research suggests that anodal tDCS may be more effective than cognitive training in preventing and slowing down the sleep-related cognitive deterioration in older adults [31]. While the benefits of tDCS on cognitive functions are widely known, it is unclear how tDCS alone and in combination with cognitive training impact sleep quality and associated functions of cognitive domains. According to our study, combining different modalities of treatments can help mild NCD-AD patients improve their subjective sleep quality and cognitive functions. Overall, our findings provide robust evidence in support of the hypothesis that baseline subjective sleep quality could predict the effectiveness of tDCS treatments with presence of greater in global cognition and working memory.

Crucially, our findings demonstrate for the first time that that treatment-induced improvements in cognitive functions and sleep quality were greater for those with poor pre-treatment sleep quality. Particularly, those who received tDCS-CCT for poor sleep quality had improved global cognition scores and greater cognitive benefits in the domain of memory function. We also observed that, in addition to efficacy, the improvements on subjective sleep quality and cognition could continue for two months (8 weeks after the treatments), and either the effect size (efficacy) or sustainability are comparable to the other non-pharmacological therapies, such as cognitive behavioral therapy (CBT) [32]. Furthermore, some studies have showed that poor sleep quality or a disrupted sleep process is associated to lower levels of cognitive abilities and plays a major role in the advancement of disease. Thus, more studies are required to determine whether the sleep changes brought on by tDCS and the ensuing cognitive enhancement reinforce the same neural pathways.

## Limitations and future directions

Our present analysis has strengths and limitations. This study brings together a number of significant findings and ideas that have long been present in the literature on ageing. The main strength of this study is, first, a modality-driven randomized clinical trial with a sizable sample size, allows us to quantify the effect of specific treatment and synergistic treatment combinations on subjective sleep quality and cognitive functions; and second,

the assessments of domain-general and domain-specific cognitive functions used in a well-characterized population diagnosed with *DSM*-5 mild NCD-AD allow us to investigate the relationship between poor sleep quality and domain-specific cognitive functions. On the other hand, although a sizeable sample size and low dropout rate were achieved, the diagnosis of mild NCD-AD was mainly based on the performance of cognitive assessments, without neuropathological backgrounds (i.e., levels of A$\beta$ and tau) or neuroimaging information (i.e., medial temporal lobe atrophy score), which may decrease the interpretation of the results. Notably, the absence of a double-dummy control treatment group (i.e., sham tDCS and controlled cognitive training) may limit the interpretation of the clinical efficacy of tDCS treatments and underestimate its potential placebo effects. With no neuroimaging information, we were unable to detect the effects of brain structures on the treatment responses. Lastly, due to the limited representation of these populations in this study, the findings might not apply to patients under the age of 60 years.

## Conclusion

According to this secondary analysis of a clinical trial, patients with mild NCD-AD who received four weeks of repeated tDCS treatment, either in a single modality or in combination, exhibited a notable improvement in their subjective sleep quality and cognitive performance. Over the course of a two-month follow-up, the tDCS recipients with poor sleep quality had more cognitive benefits and enhanced sleep quality. The status of pre-treatment subjective sleep quality is an important factor for predicting the effectiveness of tDCS treatment. These findings suggest that recurrent tDCS might be a safe and effective non-pharmacological treatment for sleep disturbances and cognitive impairment in elderly patients.

## Supporting information

**S1 Fig. The changes of PSQI total score in good and poor sleepers across time in three treatment groups.**
(JPEG)

**S2 Fig. The changes of PSQI sub scores across time in three treatment groups.**
(JPEG)

**S1 Table. Comparisons of baseline demographics and global cognition in good and poor sleepers.**
(DOCX)

**S2 Table. Treatment outcomes of the components of sleep quality (raw scores).**
(DOCX)

**S1 Checklist. CONSORT checklist.**
(PDF)

## Acknowledgments

We would like to thank all the participants and their families for their support to this study. We also thank all the research staff from the Chen Wai Wai Excise Centre for their efforts in collecting neurocognitive and clinical data. We would also like to thank Wu Wing Yin and Ng Tsz Ying from ELCHK Smart Club, and Chan Fung-man, Ms. Wong Kuk-ching and Wong Ka-wah from Christian Family Service Centre for their supports of assisting us to recruit the participants.

## Author contributions

**Conceptualization:** Hanna Lu.

**Funding acquisition:** Linda Chiu Wa Lam.

**Investigation:** Hanna Lu.

**Methodology:** Xi Ni.

**Project administration:** Hanna Lu.

**Resources:** Linda Chiu Wa Lam.

**Validation:** Hanna Lu, Xi Ni.

**Visualization:** Hanna Lu.

**Writing – original draft:** Hanna Lu.

**Writing – review & editing:** Hanna Lu, Sandra Sau Man Chan, Calvin Pak Wing Cheng, Waichi Chan, Linda Chiu Wa Lam.

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
