## [Decision Letter · Decision Letter 0]

17 May 2024

PONE-D-24-02411Pre-treatment subjective sleep quality as a predictive biomarker in preclinical Alzheimer’s disease patients treated with transcranial direct current stimulationPLOS ONE

Dear Dr. Lu,

Thank you for submitting your manuscript to PLOS ONE. After careful consideration, we feel that it has merit but does not fully meet PLOS ONE’s publication criteria as it currently stands. Therefore, we invite you to submit a revised version of the manuscript that addresses the points raised during the review process.

We look forward to receiving your revised manuscript.

Kind regards,

Xu-Qiao Chen

Academic Editor

PLOS ONE

Journal Requirements:

3. Thank you for stating the following financial disclosure: "This clinical trial was supported by the Hong Kong Research Grant Council (RGC) - General Research Fund (GRF) (GRF14108214)."  

b) If there are no restrictions, please upload the minimal anonymized data set necessary to replicate your study findings to a stable, public repository and provide us with the relevant URLs, DOIs, or accession numbers. Please see http://www.bmj.com/content/340/bmj.c181.long for guidelines on how to de-identify and prepare clinical data for publication. For a list of recommended repositories, please see https://journals.plos.org/plosone/s/recommended-repositories . You also have the option of uploading the data as Supporting Information files, but we would recommend depositing data directly to a data repository if possible.

6. Please include captions for your Supporting Information files at the end of your manuscript, and update any in-text citations to match accordingly. Please see our Supporting Information guidelines for more information: http://journals.plos.org/plosone/s/supporting-information .

Reviewers' comments:

Reviewer's Responses to Questions

**Comments to the Author**

1. Is the manuscript technically sound, and do the data support the conclusions?

Reviewer #1: Yes

Reviewer #2: No

2. Has the statistical analysis been performed appropriately and rigorously?

Reviewer #1: Yes

Reviewer #2: No

3. Have the authors made all data underlying the findings in their manuscript fully available?

Reviewer #1: Yes

Reviewer #2: Yes

4. Is the manuscript presented in an intelligible fashion and written in standard English?

Reviewer #1: Yes

Reviewer #2: No

5. Review Comments to the Author

Reviewer #1: The manuscript presents an interesting study investigating the effects of transcranial direct current stimulation (tDCS) on sleep quality and cognition in mild neurocognitive disorder due to Alzheimer’s disease (NCD-AD) patients. With some revisions to improve clarity, organization, and presentation, the manuscript could significantly enhance its impact and readability

The introduction provides a comprehensive background on the topic, but it would benefit from clearly stating the primary objectives and hypotheses. The discussion interprets the study findings well and relates them to existing literature. However, it could be structured more clearly to address each key finding separately and discuss its implications in greater detail .

The manuscript would benefit from a thorough proofreading to improve sentence structure and ensure clarity of expression.

Reviewer #2: Dear Editor,

Thank you for the opportunity to provide a review of Manuscript PONE-D-24-02411 entitled "Pre-treatment subjective sleep quality as a predictive biomarker in preclinical Alzheimer’s disease patients treated with transcranial direct current stimulation." My comments relate primarily to the adequacy of the implementation and reporting of epidemiologic and statistical procedures.

The quality of the technical English was variable. The authors must perform a thorough round of copyediting to correct the many instances of grammatical and syntactical errors in the text. These errors offered no bar to my evaluation of the manuscript.

# Incomplete reporting

The authors cannot cite previous papers when reporting key elements of the methods. For example, they state "details of randomization, masking, treatment modality, inclusion and exclusion criteria have been reported elsewhere." This paper must stand alone. The authors must provide a description of these important methodologic procedures, as is required by CONSORT.

# Identification of the primary outcome

The authors must identify the primary outcome of this manuscript and place it in the context of the trial and its protocol.

# No safety endpoints

The authors describe how safety assessments were to be conducted but fail to report any results on this outcome.

# R is not published by IBM

The authors cite R as being published by IBM. This is wrong.

# Lack of justification of correlation matrix

The authors do not justify the use of correlation techniques. What research questions were being posed here? Are these questions pre-planned? They seem opportunistic.

# DO NOT TEST significance at baseline

The authors should not perform formal statistical tests on the baseline characteristics of participants in a randomised controlled trial. This is a major error and is highly inappropriate. It sets up a statistical tautology.

Table 1's last two columns must be deleted. Any mention of statistical significance relating to Table 1 should be removed.

# Report confidence intervals

The authors should report confidence intervals. These intervals should accompany the point estimates.

# Recommendation

I cannot support the acceptance of this manuscript for publication in the Journal until the issues identified above are considered.

Thank you.

6. PLOS authors have the option to publish the peer review history of their article (what does this mean? ). If published, this will include your full peer review and any attached files.

**Do you want your identity to be public for this peer review?** For information about this choice, including consent withdrawal, please see our Privacy Policy .

Reviewer #1: No

Reviewer #2: No

While revising your submission, please upload your figure files to the Preflight Analysis and Conversion Engine (PACE) digital diagnostic tool, https://pacev2.apexcovantage.com/ . PACE helps ensure that figures meet PLOS requirements. To use PACE, you must first register as a user. Registration is free. Then, login and navigate to the UPLOAD tab, where you will find detailed instructions on how to use the tool. If you encounter any issues or have any questions when using PACE, please email PLOS at

---

## [Author Response · Author response to Decision Letter 0]

19 Dec 2024

Point-by-point responses to the reviewer’s comments on the manuscript

"Pre-treatment subjective sleep quality as a predictive biomarker of tDCS effects in preclinical Alzheimer’s disease patients: secondary analysis of a randomised clinical trial"

Comments from editors

Comment 1:

Please provide additional details regarding participant consent. In the ethics statement in the Methods and online submission information, please ensure that you have specified what type you obtained (for instance, written or verbal, and if verbal, how it was documented and witnessed). If your study included minors, state whether you obtained consent from parents or guardians. If the need for consent was waived by the ethics committee, please include this information.

Response 1:

Thank you for the suggestion. The contents have been modified as “The written informed consent was obtained from all participants or their legal guardian(s) before any assessment.” on Page 4.

Comment 2:

Thank you for stating the following financial disclosure: "This clinical trial was supported by the Hong Kong Research Grant Council (RGC) - General Research Fund (GRF) (GRF14108214)." Please state what role the funders took in the study.

Response 2:

Yes. We have stated the role of funders in the cover letter as “The funders had no role in study design, data collection and analysis, decision to publish, or preparation of the manuscript.”.

Comment 3:

Response 3:

Yes. Data Availability statement has been updated in the submission form.

Comment 4:

Your ethics statement should only appear in the Methods section of your manuscript. If your ethics statement is written in any section besides the Methods, please move it to the Methods section and delete it from any other section.

Response 4:

The “Ethics declarations” on Page 12 have been removed.

Comment 5:

Please include captions for your Supporting Information files at the end of your manuscript, and update any in-text citations to match accordingly.

Response 5:

Yes. The captions for supporting documents have been updated on Page 19.

Reviewer 1

General Comments

Comment: The manuscript presents an interesting study investigating the effects of transcranial direct current stimulation (tDCS) on sleep quality and cognition in mild neurocognitive disorder due to Alzheimer’s disease (NCD-AD) patients. With some revisions to improve clarity, organization, and presentation, the manuscript could significantly enhance its impact and readability

Response: Thank you for reviewing our paper. We also thank you for considering our study to be interesting. Based on your thoughtful and valuable comments, we have answered each of your points as follows and revised the manuscript for your further evaluation.

Specific Comment 1:

The authors state that patients with a DSM-5 diagnosis of major neurocognitive disorder were recruited and that the participants who had a history of primary bipolar or other psychotic disorders, and major neurological disorders, including stroke, transient ischemic attack or traumatic brain injury. However, the neuropathological backgrounds of the subject group are not still clear, making interpretation of the results difficult. Were they mostly Alzheimer’s disease patients? Please clarify their neurological diagnosis of the subjects in ‘Study design’ section.

Response 1:

Thank you very much for this suggestion. We totally agree with your opinion. Neuropathological backgrounds are critical for preclinical AD patients. Due to limited research funding, either structural MRI or plasma/saliva Aβ and tau testing is unavailable in this study, thus, we gave a diagnosis of mild neurocognitive disorder due to Alzheimer’s disease (NCD-AD) based on the AD-signatured impaired domains of cognition (i.e., working memory).

For better describing the neurological diagnosis of the subjects, the have been updated as “The following criteria were used to define mild NCD-AD participants: (1) evidence of modest cognitive decline in one or more cognitive domains, as indicated by a box score of ≤0.5 on the Clinical Dementia Rating (CDR) subcategory and a Cantonese Mini Mental State Examination (CMMSE) score ranging from 22 to 27; (2) no interference with daily activities that require independence; (3) with a score of episodic memory as measured by delay recall in list learning; and (4) no better explanation by other psychiatric disorders. In addition to meeting the criteria for NCD, patients with mild NCD-AD did not have uncontrolled hypertension, diabetes, or hyperlipidemia (vascular risk factors).” on Page 4. Meanwhile, we are fully acknowledged the absence of neuropathological markers in the limitations and highlighted this part as “On the other hand, although a sizeable sample size and low dropout rate were achieved, the diagnosis of mild NCD-AD was mainly based on the performance of cognitive assessments, without neuropathological backgrounds (i.e., levels of Aβ and tau) or neuroimaging information (i.e., medial temporal lobe atrophy score), which may decrease the interpretation of the results.” on Page 11.

Specific Comment 2:

The introduction provides a comprehensive background on the topic, but it would benefit from clearly stating the primary objectives and hypotheses. The discussion interprets the study findings well and relates them to existing literature. However, it could be structured more clearly to address each key finding separately and discuss its implications in greater detail.

Response 2:

Yes. In the introduction part, we highlighted our primary aim of this secondary analysis as “Therefore, in order to make significant progress in comprehending the reciprocal relationship between subjective sleep quality and cognitive functions, our primary objective was to present a thorough picture of subjective sleep quality, global cognition and domain-specific cognitive functions related to tDCS treatments. The secondary objective was to identify the links between pre-treatment subjective sleep quality and the treatment-induced cognitive changes.” on page 4. The discuss parts have been structured and separated with sub-titles.

Specific Comment 3:

The manuscript would benefit from thorough proofreading to improve sentence structure and ensure clarity of expression.

Response 3: Thank you very much for the suggestion. We have modified some of the descriptions for clearly explaining our viewpoints. The proofreading of this revised manuscript has been done.

Reviewer 2

General Comments

Comment: Thank you for the opportunity to provide a review of Manuscript PONE-D-24-02411 entitled "Pre-treatment subjective sleep quality as a predictive biomarker in preclinical Alzheimer’s disease patients treated with transcranial direct current stimulation." My comments relate primarily to the adequacy of the implementation and reporting of epidemiologic and statistical procedures. The quality of the technical English was variable.

The authors must perform a thorough round of copyediting to correct the many instances of grammatical and syntactical errors in the text. These errors offered no bar to my evaluation of the manuscript.

Response: Thank you very much for taking the time to review our manuscript. The detailed comments are very thoughtful and helpful. In this revision, we have done thorough proofreading to improve the quality of this manuscript.

Major Concerns

Specific Comment 1:

The authors cannot cite previous papers when reporting key elements of the methods. For example, they state "details of randomization, masking, treatment modality, inclusion and exclusion criteria have been reported elsewhere." This paper must stand alone. The authors must provide a description of these important methodologic procedures, as is required by CONSORT.

Response 1:

Thank you for your suggestion. We have updated the inclusion criteria of mild neurocognitive disorder due to Alzheimer’s disease (NCD-AD) on Page 4. The sentence of “The details of randomization, masking, treatment modality, inclusion and exclusion criteria have been reported elsewhere.” has been removed from the main contents.

Specific Comment 2:

The authors must identify the primary outcome of this manuscript and place it in the context of the trial and its protocol.

Response 2:

Thank you for your comment. We apologize for any confusion that we did not explain the contents clearly in this paper. This study is a secondary analysis of our previous randomized clinical trial. Thus, for clearly presenting the primary aims of this paper, we modified the title as “Pre-treatment subjective sleep quality as a predictive biomarker of tDCS effects in preclinical Alzheimer’s disease patients: secondary analysis of a randomised clinical trial”. Meanwhile, we highlighted the study aims as “In order to make significant progress in comprehending the reciprocal relationship between subjective sleep quality and cognitive functions, our primary objective was to present a thorough picture of subjective sleep quality, global cognition and domain-specific cognitive functions related to tDCS treatments. The secondary objective was to identify the links between pre-treatment subjective sleep quality and the treatment-induced cognitive changes.” on Page 4

Specific Comment 3:

The authors describe how safety assessments were to be conducted but fail to report any results on this outcome.

Response 3:

Thank you for this suggestion. We have reported the adverse events and safety on Page 8 as “All patients reported that the treatments were well-tolerated following twelve sessions of treatments. In tDCS groups, the most significant adverse events were mild skin injury under cathodal electrode in three patients. However, the above skin injuries were recovered after one week. None of the patients reported other adverse events, such as nausea, seizures, headache, or dizziness.”.

Specific Comment 4:

The authors cite R as being published by IBM. This is wrong.

Response 4:

Thank you for pointing out this issue. We apologize for the confusion. The sentence has been revised as “The computations were performed using R package.” on Page 6.

Specific Comment 5:

The authors do not justify the use of correlation techniques. What research questions were being posed here? Are these questions pre-planned? They seem opportunistic.

Response 5:

Yes. Thank you very much for this comment. We apologize for the confusion. In this revision, we highlight the aim of this study as “The secondary objective was to identify the links between pre-treatment subjective sleep quality and the treatment-induced cognitive changes.” on Page 4. Thus, we examined the correlation matrices in the three treatment groups.

Specific Comment 6:

The authors should not perform formal statistical tests on the baseline characteristics of participants in a randomised controlled trial. This is a major error and is highly inappropriate. It sets up a statistical tautology. Table 1's last two columns must be deleted. Any mention of statistical significance relating to Table 1 should be removed.

Response 6:

Thank you for this comment. The last two columns in table 1 have been removed.

---

## [Decision Letter · Decision Letter 1]

27 Dec 2024

PONE-D-24-02411R1Pre-treatment subjective sleep quality as a predictive biomarker of tDCS effects in preclinical Alzheimer’s disease patients: secondary analysis of a randomised clinical trialPLOS ONE

Dear Dr. Lu,

Thank you for submitting your manuscript to PLOS ONE. After careful consideration, we feel that it has merit but does not fully meet PLOS ONE’s publication criteria as it currently stands. Therefore, we invite you to submit a revised version of the manuscript that addresses the points raised during the review process.

The authors have addressed all the comments raised by the reviewers, but the English needs to be refined before acceptance.

We look forward to receiving your revised manuscript.

Kind regards,

Xu-Qiao Chen

Academic Editor

PLOS ONE

Journal Requirements:

Additional Editor Comments:

Kindly follow the reviewer's recommendations to refine the English language.

Reviewers' comments:

Reviewer's Responses to Questions

**Comments to the Author**

1. If the authors have adequately addressed your comments raised in a previous round of review and you feel that this manuscript is now acceptable for publication, you may indicate that here to bypass the “Comments to the Author” section, enter your conflict of interest statement in the “Confidential to Editor” section, and submit your "Accept" recommendation.

Reviewer #2: (No Response)

2. Is the manuscript technically sound, and do the data support the conclusions?

Reviewer #2: Yes

3. Has the statistical analysis been performed appropriately and rigorously?

Reviewer #2: Yes

4. Have the authors made all data underlying the findings in their manuscript fully available?

Reviewer #2: Yes

5. Is the manuscript presented in an intelligible fashion and written in standard English?

Reviewer #2: No

6. Review Comments to the Author

Reviewer #2: Dear Editor,

Thank you for the opportunity to provide a review of Manuscript PONE-D-24-02411R1 entitled "Pre-treatment subjective sleep quality as a predictive biomarker of tDCS effects in preclinical Alzheimer’s disease patients: secondary analysis of a randomised clinical trial." My comments relate primarily to the adequacy of the implementation and reporting of epidemiologic and statistical procedures.

I understand that the manuscript underwent a previous round of peer-review.

The quality of the technical English remains variable, despite the authors' attempt at identifying and correcting them. The authors must perform another round of copyediting to correct grammatical and syntactical errors in the text. I strongly suggest that they engage the services of a native English language speaker.

I am satisfied that the issues I raised in the previous round were addressed satisfactorily.

Thank you.

7. PLOS authors have the option to publish the peer review history of their article (what does this mean? ). If published, this will include your full peer review and any attached files.

**Do you want your identity to be public for this peer review?** For information about this choice, including consent withdrawal, please see our Privacy Policy .

Reviewer #2: No

While revising your submission, please upload your figure files to the Preflight Analysis and Conversion Engine (PACE) digital diagnostic tool, https://pacev2.apexcovantage.com/

---

## [Author Response · Author response to Decision Letter 1]

19 Dec 2024

Point-by-point responses to the reviewer’s comments on the manuscript

"Pre-treatment subjective sleep quality as a predictive biomarker of tDCS effects in preclinical Alzheimer’s disease patients: secondary analysis of a randomised clinical trial"

Comments from editors

Comment 1:

Please provide additional details regarding participant consent. In the ethics statement in the Methods and online submission information, please ensure that you have specified what type you obtained (for instance, written or verbal, and if verbal, how it was documented and witnessed). If your study included minors, state whether you obtained consent from parents or guardians. If the need for consent was waived by the ethics committee, please include this information.

Response 1:

Thank you for the suggestion. The contents have been modified as “The written informed consent was obtained from all participants or their legal guardian(s) before any assessment.” on Page 4.

Comment 2:

Thank you for stating the following financial disclosure: "This clinical trial was supported by the Hong Kong Research Grant Council (RGC) - General Research Fund (GRF) (GRF14108214)." Please state what role the funders took in the study.

Response 2:

Yes. We have stated the role of funders in the cover letter as “The funders had no role in study design, data collection and analysis, decision to publish, or preparation of the manuscript.”.

Comment 3:

Response 3:

Yes. Data Availability statement has been updated in the submission form.

Comment 4:

Your ethics statement should only appear in the Methods section of your manuscript. If your ethics statement is written in any section besides the Methods, please move it to the Methods section and delete it from any other section.

Response 4:

The “Ethics declarations” on Page 12 have been removed.

Comment 5:

Please include captions for your Supporting Information files at the end of your manuscript, and update any in-text citations to match accordingly.

Response 5:

Yes. The captions for supporting documents have been updated on Page 19.

Reviewer 1

General Comments

Comment: The manuscript presents an interesting study investigating the effects of transcranial direct current stimulation (tDCS) on sleep quality and cognition in mild neurocognitive disorder due to Alzheimer’s disease (NCD-AD) patients. With some revisions to improve clarity, organization, and presentation, the manuscript could significantly enhance its impact and readability

Response: Thank you for reviewing our paper. We also thank you for considering our study to be interesting. Based on your thoughtful and valuable comments, we have answered each of your points as follows and revised the manuscript for your further evaluation.

Specific Comment 1:

The authors state that patients with a DSM-5 diagnosis of major neurocognitive disorder were recruited and that the participants who had a history of primary bipolar or other psychotic disorders, and major neurological disorders, including stroke, transient ischemic attack or traumatic brain injury. However, the neuropathological backgrounds of the subject group are not still clear, making interpretation of the results difficult. Were they mostly Alzheimer’s disease patients? Please clarify their neurological diagnosis of the subjects in ‘Study design’ section.

Response 1:

Thank you very much for this suggestion. We totally agree with your opinion. Neuropathological backgrounds are critical for preclinical AD patients. Due to limited research funding, either structural MRI or plasma/saliva Aβ and tau testing is unavailable in this study, thus, we gave a diagnosis of mild neurocognitive disorder due to Alzheimer’s disease (NCD-AD) based on the AD-signatured impaired domains of cognition (i.e., working memory).

For better describing the neurological diagnosis of the subjects, the have been updated as “The following criteria were used to define mild NCD-AD participants: (1) evidence of modest cognitive decline in one or more cognitive domains, as indicated by a box score of ≤0.5 on the Clinical Dementia Rating (CDR) subcategory and a Cantonese Mini Mental State Examination (CMMSE) score ranging from 22 to 27; (2) no interference with daily activities that require independence; (3) with a score of episodic memory as measured by delay recall in list learning; and (4) no better explanation by other psychiatric disorders. In addition to meeting the criteria for NCD, patients with mild NCD-AD did not have uncontrolled hypertension, diabetes, or hyperlipidemia (vascular risk factors).” on Page 4. Meanwhile, we are fully acknowledged the absence of neuropathological markers in the limitations and highlighted this part as “On the other hand, although a sizeable sample size and low dropout rate were achieved, the diagnosis of mild NCD-AD was mainly based on the performance of cognitive assessments, without neuropathological backgrounds (i.e., levels of Aβ and tau) or neuroimaging information (i.e., medial temporal lobe atrophy score), which may decrease the interpretation of the results.” on Page 11.

Specific Comment 2:

The introduction provides a comprehensive background on the topic, but it would benefit from clearly stating the primary objectives and hypotheses. The discussion interprets the study findings well and relates them to existing literature. However, it could be structured more clearly to address each key finding separately and discuss its implications in greater detail.

Response 2:

Yes. In the introduction part, we highlighted our primary aim of this secondary analysis as “Therefore, in order to make significant progress in comprehending the reciprocal relationship between subjective sleep quality and cognitive functions, our primary objective was to present a thorough picture of subjective sleep quality, global cognition and domain-specific cognitive functions related to tDCS treatments. The secondary objective was to identify the links between pre-treatment subjective sleep quality and the treatment-induced cognitive changes.” on page 4. The discuss parts have been structured and separated with sub-titles.

Specific Comment 3:

The manuscript would benefit from thorough proofreading to improve sentence structure and ensure clarity of expression.

Response 3: Thank you very much for the suggestion. We have modified some of the descriptions for clearly explaining our viewpoints. The proofreading of this revised manuscript has been done.

Reviewer 2

General Comments

Comment: Thank you for the opportunity to provide a review of Manuscript PONE-D-24-02411 entitled "Pre-treatment subjective sleep quality as a predictive biomarker in preclinical Alzheimer’s disease patients treated with transcranial direct current stimulation." My comments relate primarily to the adequacy of the implementation and reporting of epidemiologic and statistical procedures. The quality of the technical English was variable.

The authors must perform a thorough round of copyediting to correct the many instances of grammatical and syntactical errors in the text. These errors offered no bar to my evaluation of the manuscript.

Response: Thank you very much for taking the time to review our manuscript. The detailed comments are very thoughtful and helpful. In this revision, we have done thorough proofreading to improve the quality of this manuscript.

Major Concerns

Specific Comment 1:

The authors cannot cite previous papers when reporting key elements of the methods. For example, they state "details of randomization, masking, treatment modality, inclusion and exclusion criteria have been reported elsewhere." This paper must stand alone. The authors must provide a description of these important methodologic procedures, as is required by CONSORT.

Response 1:

Thank you for your suggestion. We have updated the inclusion criteria of mild neurocognitive disorder due to Alzheimer’s disease (NCD-AD) on Page 4. The sentence of “The details of randomization, masking, treatment modality, inclusion and exclusion criteria have been reported elsewhere.” has been removed from the main contents.

Specific Comment 2:

The authors must identify the primary outcome of this manuscript and place it in the context of the trial and its protocol.

Response 2:

Thank you for your comment. We apologize for any confusion that we did not explain the contents clearly in this paper. This study is a secondary analysis of our previous randomized clinical trial. Thus, for clearly presenting the primary aims of this paper, we modified the title as “Pre-treatment subjective sleep quality as a predictive biomarker of tDCS effects in preclinical Alzheimer’s disease patients: secondary analysis of a randomised clinical trial”. Meanwhile, we highlighted the study aims as “In order to make significant progress in comprehending the reciprocal relationship between subjective sleep quality and cognitive functions, our primary objective was to present a thorough picture of subjective sleep quality, global cognition and domain-specific cognitive functions related to tDCS treatments. The secondary objective was to identify the links between pre-treatment subjective sleep quality and the treatment-induced cognitive changes.” on Page 4

Specific Comment 3:

The authors describe how safety assessments were to be conducted but fail to report any results on this outcome.

Response 3:

Thank you for this suggestion. We have reported the adverse events and safety on Page 8 as “All patients reported that the treatments were well-tolerated following twelve sessions of treatments. In tDCS groups, the most significant adverse events were mild skin injury under cathodal electrode in three patients. However, the above skin injuries were recovered after one week. None of the patients reported other adverse events, such as nausea, seizures, headache, or dizziness.”.

Specific Comment 4:

The authors cite R as being published by IBM. This is wrong.

Response 4:

Thank you for pointing out this issue. We apologize for the confusion. The sentence has been revised as “The computations were performed using R package.” on Page 6.

Specific Comment 5:

The authors do not justify the use of correlation techniques. What research questions were being posed here? Are these questions pre-planned? They seem opportunistic.

Response 5:

Yes. Thank you very much for this comment. We apologize for the confusion. In this revision, we highlight the aim of this study as “The secondary objective was to identify the links between pre-treatment subjective sleep quality and the treatment-induced cognitive changes.” on Page 4. Thus, we examined the correlation matrices in the three treatment groups.

Specific Comment 6:

The authors should not perform formal statistical tests on the baseline characteristics of participants in a randomised controlled trial. This is a major error and is highly inappropriate. It sets up a statistical tautology. Table 1's last two columns must be deleted. Any mention of statistical significance relating to Table 1 should be removed.

Response 6:

Thank you for this comment. The last two columns in table 1 have been removed.

---

## [Decision Letter · Decision Letter 2]

3 Jan 2025

Pre-treatment subjective sleep quality as a predictive biomarker of tDCS effects in preclinical Alzheimer’s disease patients: secondary analysis of a randomised clinical trial

PONE-D-24-02411R2

Dear Dr. Lu,

We’re pleased to inform you that your manuscript has been judged scientifically suitable for publication and will be formally accepted for publication once it meets all outstanding technical requirements.

Kind regards,

Xu-Qiao Chen

Academic Editor

PLOS ONE

Additional Editor Comments (optional):

Reviewers' comments:

Reviewer's Responses to Questions

**Comments to the Author**

1. If the authors have adequately addressed your comments raised in a previous round of review and you feel that this manuscript is now acceptable for publication, you may indicate that here to bypass the “Comments to the Author” section, enter your conflict of interest statement in the “Confidential to Editor” section, and submit your "Accept" recommendation.

Reviewer #2: (No Response)

2. Is the manuscript technically sound, and do the data support the conclusions?

Reviewer #2: Yes

3. Has the statistical analysis been performed appropriately and rigorously?

Reviewer #2: Yes

4. Have the authors made all data underlying the findings in their manuscript fully available?

Reviewer #2: Yes

5. Is the manuscript presented in an intelligible fashion and written in standard English?

Reviewer #2: No

6. Review Comments to the Author

Reviewer #2: See response to the editor. See response to the editor. See response to the editor. See response to the editor.

7. PLOS authors have the option to publish the peer review history of their article (what does this mean? ). If published, this will include your full peer review and any attached files.

**Do you want your identity to be public for this peer review?** For information about this choice, including consent withdrawal, please see our Privacy Policy .

Reviewer #2: No

---

## [Editor Report · Acceptance letter]

PONE-D-24-02411R2

PLOS ONE

Dear Dr. Lu,

I'm pleased to inform you that your manuscript has been deemed suitable for publication in PLOS ONE. Congratulations! Your manuscript is now being handed over to our production team.

Kind regards,

on behalf of

Dr. Xu-Qiao Chen

Academic Editor

PLOS ONE